# What matters to women in the postnatal period: A meta-synthesis of qualitative studies

**Kenneth Finlayson**[1]ⓘ[☯]*, **Nicola Crossland**[2☯], **Mercedes Bonet**[3‡], **Soo Downe**ⓘ[1‡]

**1** Research in Childbirth and Health (ReaCH) group, University of Central Lancashire, Preston, United Kingdom, **2** Faculty of Health and Wellbeing, University of Central Lancashire, Preston, United Kingdom, **3** Department of Reproductive Health and Research, UNDP/UNFPA/UNICEF/WHO/World Bank Special Programme of Research, Development and Research Training in Human Reproduction (HRP), World Health Organization, Geneva, Switzerland

☯ These authors contributed equally to this work.
‡ These authors also contributed equally to this work.
* kwfinlayson1@uclan.ac.uk

## Abstract

### Introduction

The postnatal period is an underserved aspect of maternity care. Guidelines for postnatal care are not usually informed by what matters to the women who use it. This qualitative systematic review was undertaken to identify what matters to women in the postnatal period, to inform the scope of a new World Health Organization (WHO) postnatal guideline.

### Methods

We searched MEDLINE, CINAHL, PsycINFO, POPLINE, Global Index Medicus, EMBASE, LILACS, AJOL, and reference lists of eligible studies published January 2000–July 2019, reporting qualitative data on women's beliefs, expectations, and values relating to the postnatal period.

### Data collection and analysis

Author findings were extracted, coded and synthesised using techniques derived from thematic synthesis. Confidence in the quality, coherence, relevance and adequacy of data underpinning the resulting findings was assessed using GRADE-CERQual.

### Results

We included 36 studies from 15 countries, representing the views of more than 800 women. Confidence in most results was moderate to high. What mattered to women was a positive postnatal experience where they were able to adapt to their new self-identity and develop a sense of confidence and competence as a mother; adjust to changes in their intimate and family relationships, including their relationship to their baby; navigate ordinary physical and emotional challenges; and experience the dynamic achievement of personal growth as they adjust to the 'new normal' of motherhood and parenting in their own cultural context.

**Data Availability Statement:** All relevant data are within the manuscript and its Supporting Information files.

**Funding:** The work was commissioned to the University of Central Lancashire by the UNDP/

UNFPA/UNICEF/WHO/World Bank Special Programme of Research, Development and Research Training in Human Reproduction (HRP), a cosponsored program executed by the World Health Organization (WHO). The funders had no role in study design, data collection and analysis, decision to publish, or preparation of the manuscript.

**Competing interests:** The authors have declared that no competing interests exist.

## Conclusion

This review provides evidence that what matters to women in the postnatal period is achieving positive motherhood (including maternal self-esteem, competence, and autonomy), as well as fulfilling adaptation to changed intimate and family relationships, and (re)gaining health and wellbeing for both their baby, and themselves. Where this process is optimal, it also results in joy, self-confidence, and an enhanced capacity to thrive in the new integrated identity of 'woman and mother'.

## Introduction

The postnatal period is a significant phase in the lives of mothers and babies. It is a time of adaptation to parenthood, of the development of secure attachment for the neonate and young infant, and a time where bonds can develop within the family and with the community [1]. For some mothers and babies it includes ill-health [1,2]. Specific maternal mortality/morbidity data relating to the postnatal phase is limited; however, recent figures indicate there are an estimated 303 000 maternal deaths annually resulting from complications related to pregnancy, childbirth or the postnatal period [2]. The majority of these deaths occur postnatally, with post-partum haemorrhage (PPH) the most common cause of maternal death [3]. Neonatal data are more widely available and recent estimates indicate there are almost three million neonatal deaths (deaths in the first 28 days after birth) each year, most of which are preventable [4]. The Global Strategy for Women's, Children's and Adolescents' Health 2016–2030 highlights the importance of postnatal care for mothers and babies in ending preventable deaths and ensuring health and wellbeing [5]. Strategies designed to reduce the rates of maternal and neonatal mortality are also endorsed by the World Health Organization and emphasize the importance of the postnatal period in achieving this goal [6,7].

By definition the postnatal period is the phase of life immediately following childbirth. Its duration is culturally variable, but the first six weeks after childbirth is common cross-culturally, and the WHO defines the postnatal phase as beginning immediately after the birth of the baby and extending for up to six weeks (42 days) after birth [1]. In terms of care provision, the postnatal period tends to be divided into the immediate (first 24h), early (days 2–7) and late (days 8–42) periods. Postnatal care in the immediate phase is likely to be facility based in many settings, and focused on key clinical indicators for the baby and monitoring of general wellbeing for the mother [8,9]. Early and late postnatal care is more likely to be community based and focused on maximizing maternal and newborn health and wellbeing. Postnatal contacts provide an opportunity for healthcare providers to facilitate healthy breastfeeding practices, screen for postpartum depression, monitor the baby's growth and overall health status, treat childbirth-related complications, counsel women about their family planning options and refer the mother and baby for specialized care if necessary [8,9].

To achieve the aim of both thriving and flourishing, as well as surviving, it is important that mothers and families are supported and enabled to experience the optimal start in life with their newborn [1]. However, the postnatal period is a neglected phase of maternity care with more emphasis and resources placed on antenatal and intrapartum care [10]. According to the recent 'Countdown to 2030' report postnatal services have the lowest median national coverage of interventions on the continuum of maternal and child healthcare [11]. In addition, utilization of postnatal services varies widely, particularly in low and middle-income countries (LMICs), where barriers to access based largely on socio-economic circumstances and rurality

limit engagement [12]. Innovative approaches like m-health have shown some promise in raising awareness and increasing coverage in some LMIC settings [13] and giving birth in a recognized health facility may enhance further engagement with postnatal services [14]. However, the situation in many under resourced LMICs is hugely variable and, even in hospital settings, many women fail to receive the most basic pre-discharge check-up [15,16].

Existing literature focusses predominately on the effectiveness of specific postnatal interventions, or around women's experiences of postnatal care services (see for example, [17–19]). This has resulted in limited information about what women themselves value during this period. The aim of the review is, therefore, to identify what matters to women in the postnatal period, in order to better understand how postnatal services can be optimally designed to deliver a positive experience to meet the needs of women, their families and their neonates.

## Methods

We conducted a thematic synthesis of qualitative studies in accordance with the PRISMA guidelines (See S1 Table). We included studies where the focus was on healthy women and neonates, irrespective of parity, mode of birth or place of birth. Study assessment included the use of a validated quality appraisal tool [20]. Thematic synthesis techniques [21] were used for analysis and synthesis, and GRADE-CERQual [22] was applied to the resulting review findings.

### Criteria for inclusion

The focus of the review was what matters to women in the postnatal period. Studies solely investigating women's views and experiences of postnatal care provision were excluded. We included qualitative studies reporting first-hand accounts (including the views, expectations and perceptions) of women of any parity who gave birth in any setting (including in a health facility or at home). Studies published before 2000 were excluded to ensure that the data reflect the views of the most recent generation of women of childbearing age. Studies solely investigating experiences of, or support for, breastfeeding or infant feeding, or special neonatal care, or any other specific conditions or circumstances were excluded. No language restriction was imposed. Case studies, conference abstracts and unpublished PhD or Masters theses were not included.

### Reflexive note

Quality standards for rigor in qualitative research [20] require that authors consider how their views and opinions could influence decisions made in the design and conduct of a study, and how emerging findings influence those views and opinions. KF, NC and MB believe that the postnatal period is of short and long-term importance for maternal and child health and wellbeing, and is currently underserved by health services and systems. SD has experience of providing postnatal care as a midwife in the UK, and of receiving it as a mother. She is aware of the lack of support experienced by many mothers at this time. From her work on metasynthesis reviews of what matters to women in the antenatal period, she expects to find that adaptation to motherhood will feature in what matters to women in the postnatal period. We consciously sought disconfirming data for these prior beliefs, to ensure that we did not over-emphasise data that reinforced our existing positions.

### Search strategy

A search strategy was developed using a PEO (Populations, Exposure, Outcome) structure with the addition of searches to identify qualitative studies. Systematic searches were carried

out in May 2019 in CINAHL, MEDLINE, PsycINFO, EMBASE, Global Index Medicus, POPLINE, African Journals Online and LILACS. Searches were carried out using keywords for the Population, Exposure, Outcomes, and methodology where possible, or for smaller databases, using exposure keywords only. An example search strategy is shown in Box 1.

---

### Box 1. Example search strategy (MEDLINE)

1. Postnatal Care/

2. Qualitative Research/

3. (woman or women$ or mother$ or mum$ or mom$).ti,ab.

4. (qualitative or interview$ or focus group$ or ethnograph$ or phenomenol$ or mixed methods or grounded theory).ti,ab.

5. 2 or 4

6. (view$ or or expectation or perspective$ or perception$ or opinion$ or belief$ or understand$ or encounter$ or attitude$ or prefer$ or provision or feel$ or think or thought$ or value$).ti,ab.

7. (postnatal or postpartum or puerperium or puerperal or afterbirth or lying in or confinement).ti,ab.

8. 1 or 7

9. 3 and 5 and 6 and 8

10. limit 9 to yr = "2000 -Current"

---

### Study selection

We collated records into Endnote X8.2, removed duplicates, and where possible, removed irrelevant records based on title. The remaining records were then assessed by abstract. To check for consistency, two review authors (KF, NC) independently screened the same subset of abstracts against the a priori inclusion/exclusion criteria, and each author then screened half the remaining abstracts, discarding irrelevant records. Two review authors (KF, NC) assessed the full texts of papers, with adjudication by a third author (SD or MB) and agreed on the final list of included studies. The full texts of studies published in languages other than English were translated into English using freely available online software (Google Translate).

### Quality assessment

Included studies were appraised using an instrument developed by Walsh and Downe [20] and modified by Downe et al [23]. Studies were rated against 11 pre-defined criteria, and then allocated a score from A–D, where A represented a study with no, or few flaws, with high credibility, transferability, dependability and confirmability; B, a study with some flaws, unlikely to affect the credibility, transferability, dependability and/or confirmability of the study; C, a study with some flaws that may affect the credibility, transferability, dependability and/or confirmability of the study; and D, a study with significant flaws that are very likely to affect the credibility, transferability, dependability and/or confirmability of the study. Studies were

appraised by two authors independently (KF, NC) and a 20% sample were cross checked by the same authors to ensure consistency. Any studies where there were scoring discrepancies of more than a grade were referred to another author (SD) for moderation. Studies scoring C or higher were included in the final analysis.

### Data extraction and analysis

The analytic process broadly followed the principles of thematic synthesis [21]. We started by reading the papers closely and identified an index paper that best reflected the focus of the review. The themes and findings identified by the authors of the index paper were coded and entered onto a spreadsheet to develop an initial thematic framework. The findings of all the remaining papers were coded and mapped to this framework, which continued to develop as the data from each paper were added (see S1 Appendix). This process includes looking for what is similar between papers and for what contradicts ('disconfirms') the emerging themes. For the disconfirming process we consciously looked for data that would contradict our emerging themes, or our prior beliefs and views related to the topic of the review. Data extraction and synthesis proceeded concurrently. Descriptive themes were developed from the quote material and author interpretations of the studies.

Once the framework of descriptive review findings was agreed by all of the authors, the level of confidence in each review finding was assessed using the GRADE-CERQual tool [22] and agreed by consensus between two review authors (KF, NC). GRADE-CERQual assesses the methodological limitations and relevance to the review of the studies contributing to a review finding, the coherence of the review finding, and the adequacy of data supporting a review finding. Based on these criteria, review findings were graded for confidence using a classification system ranging from 'high' to 'moderate' to 'low' to 'very low'. The data extraction, analysis, synthesis and CERQual grading stages are illustrated in the S1 Appendix. Following CERQual assessment the review findings were grouped into higher order, analytical themes and the final framework was agreed by consensus amongst the authors. A summary statement encapsulating all of the findings was developed to provide an overall conceptual proposition to give insight into what women want in the postnatal period. Key concepts relating to a positive postnatal experience were derived from the summary statement.

## Results

### Included studies

Systematic searches yielded 2920 records. Ninety-five duplicate records were removed. Of the remaining records, 2473 were excluded by title and a further 213 excluded by abstract, leaving 139 full-text papers to be screened. Thirty-seven full-text papers were quality appraised and one [24] was excluded following translation because it focused on women's experiences of postnatal care rather than their needs and expectations, meaning that 36 were included. No papers were excluded on quality grounds, and after our quality appraisal checks 8 included papers were rated A, 16 were rated B, and 12 were rated C. Thirty-six papers were therefore included in the data analysis [25–60]. A PRISMA flow chart illustrating the selection process is shown in Fig 1.

The characteristics and quality appraisal ratings of the included papers are summarised in Table 1. Papers were published between 2003 and 2019 and 33/36 were from high or upper-middle income countries (according to World Bank, 2019). By country the papers were from Australia (n = 4), Belgium (n = 1), Brazil (n = 7), Canada (n = 3), Iran (n = 1), Ireland (n = 1), Kenya (n = 1), Malawi (n = 1), Norway (n = 1), Spain (n = 2), Switzerland (n = 4),Taiwan (n = 2), Tanzania (n = 1), UK (n = 4), USA (n = 3). Two papers from Switzerland [35,36]

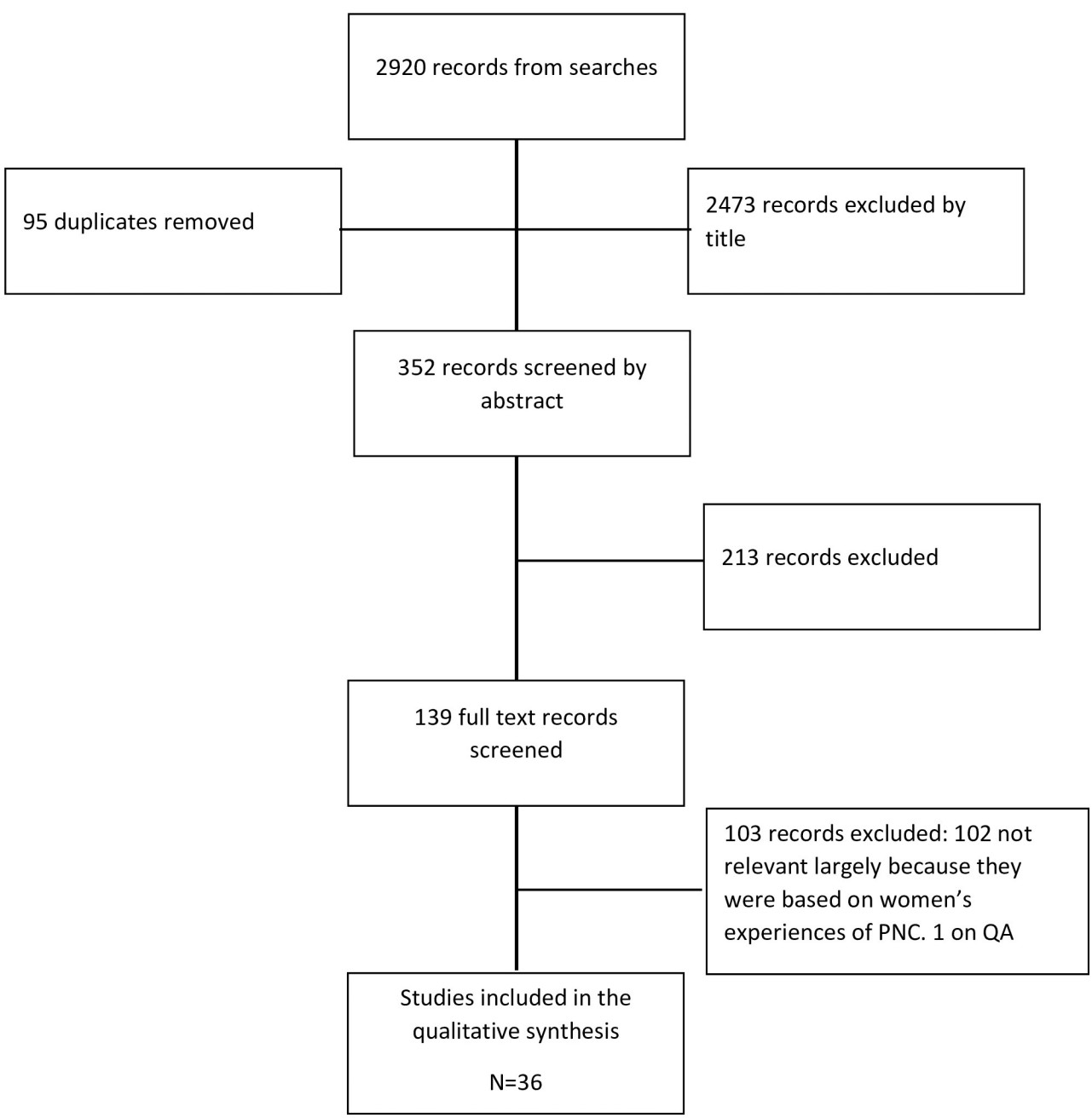

**Fig 1. PRISMA flowchart.**

appeared to report findings from the same study, and three papers from Brazil [50–52] also appeared to report findings from the same study; all papers were included in the analysis as they reported on different themes. Two papers were translated from Portuguese into English [42,52] and one from Spanish to English [44] using a recognized software tool ('Google Translate'). Nineteen of the thirty-six included studies were focused on the views of first-time mothers only and the remaining seventeen incorporated the views of women with a range of parities. Most data were collected by individual interviews and/or focus or discussion groups. The studies incorporated a variety of methodological approaches including phenomenological

**Table 1. Characteristics of included studies.**

| Author(s) and Study | Year | Country | Resource | Context | Study design | Participants | Quality Rating |
|---|---|---|---|---|---|---|---|
| Clark A, Skouteris H, Wertheim EH, Paxton SJ, Milgrom J. *My baby body: A qualitative insight into women's body-related experiences and mood during pregnancy and the postpartum.* [25] | 2009 | Australia | High | Urban hospitals in two cities in Australia | Qualitative and descriptive informed by interviews | 20 perinatal women of various parities (10 interviewed in late pregnancy & 10, 5–12 weeks postpartum) | B+ |
| Cronin C, McCarthy G. *First-time mothers–identifying their needs, perceptions and experiences.* [26] | 2003 | Ireland | High | Urban context in South Ireland | Qualitative and descriptive informed by interviews and focus groups | 13 postnatal women interviewed up to 9 months postpartum | C+ |
| Deave T, Johnson D, Ingram J. *Transition to parenthood: the needs of parents in pregnancy and early parenthood.* [27] | 2008 | UK | High | Urban. Two healthcare settings in SW England | Qualitative and descriptive informed by interviews | 24 nulliparous women interviewed at 3–4 months postpartum | B |
| Demarchi RF, Nascimento VF, Borges AP, Terças ACP, Grein TAD, Baggio É. *Perception of pregnant women and primiparous puerpera's on maternity.* [28] | 2017 | Brazil | Upper Middle | Urban. Eight clinics in the north of Brazil | Qualitative and descriptive incorporating Peplau's theory of interpersonal relationships and informed by interviews | 11 primiparous women interviewed during late pregnancy and again between 11 & 45 days postpartum. | B+ |
| DeMaria AL, Delay C, Sundstrom B, Wakefield AL, Avina A, Meier S. *Understanding women's postpartum sexual experiences.* [29] | 2019 | USA | High | Urban centres in South Carolina | Qualitative and descriptive informed by interviews | 70 women of various ages and parities interviewed in a historical context | B- |
| Forster DA, McLachlan HL, Rayner J, Yelland J, Gold L, Rayner S. *The early postnatal period: exploring women's views, expectations and experiences of care using focus groups in Victoria, Australia.* [30] | 2008 | Australia | High | Two urban hospitals in the state of Victoria | Qualitative and descriptive informed largely by focus groups (4 interviews) | Total of 52 people in 8 focus groups:- 42 postpartum women (up to 12 months), 8 pregnant women and 2 partners | B- |
| Gaboury J, Capaday S, Somera J, Purden M. *Effect of the Postpartum Hospital Environment on the Attainment of Mothers' and Fathers' Goals.* [31] | 2017 | Canada | High | Urban hospital—participants recruited from postnatal ward | Qualitative and descriptive informed by interviews | 10 women (of various parities) and 8 partners interviewed separately in the immediate post-partum phase (within 48 hours) | B+ |
| Haga SM, Lynne A, Slinning K, Kraft P. *A qualitative study of depressive symptoms and wellbeing among first-time mothers.* [32] | 2102 | Norway | High | Urban clinics in Oslo | Qualitative and descriptive informed by interviews | 12 primiparous women who had given birth within the previous 12 months | C+ |
| Javadifar N, Majlesi; F, Nikbakht A, Nedjat S, Montazeri A. *Journey to Motherhood in the First Year After Child Birth.* [33] | 2016 | Iran | Upper Middle | Urban clinics in Tehran and Ahwaz | Qualitative and descriptive informed by interviews | 26 primiparous women who had given birth within the previous 12 months | B |
| Kalinowski LC, Favero L, Carraro TE, Wall ML, Lacerda MR. *Postpartum primipara at home and associated nursing care: a data-based theory.* [34] | 2012 | Brazil | Upper Middle | Urban—in the homes of women living in a Metropolitan area (Curitiba) | Qualitative and exploratory using Grounded Theory approach informed by interviews | 16 primiparous women, 6 of whom interviewed within the first 10 days; 6 between 11 and 42 days and 4 beyond 42 days | C+ |

*(Continued)*

**Table 1.** (Continued)

| Author(s) and Study | Year | Country | Resource | Context | Study design | Participants | Quality Rating |
|---|---|---|---|---|---|---|---|
| Kurth E, Krähenbühl K, Eicher M, Rodmann S, Folmli L, Conzelmann C, et al. *Safe start at home: what parents of newborns need after early discharge from hospital—a focus group study*. [35] | 2016 | Switzerland | High | Urban and sub-urban locations in the region of Basel | Qualitative and descriptive using a 'playful' design informed by focus groups | 24 participants in 6 focus groups including 1 with partners (n = 4) and 1 with Turkish speaking women (n = 2) conducted up to 9 months postpartum | B |
| Kurth E, Powell Kennedy H, Stutz EZ, Kesselring A, Fornaro I, Spichiger E. *Responding to a crying infant–You do not learn it overnight: A phenomenological study*. [36] | 2014 | Switzerland | High | Urban—postnatal ward of a tertiary hospital in Basel | Longitudinal qualitative study utilizing an interpretive phenomenological approach | 15 mothers of various parities including participant observations in the postnatal ward and two narrative interviews at participants' homes at 6–8 and 12–14 weeks post-partum | A- |
| Kurth E, Spichiger E, Stutz EZ, Biedermann J, Hosli I, Kennedy HP. *Crying babies, tired mothers—challenges of the postnatal hospital stay: an interpretive phenomenological study*. [37] | 2010 | Switzerland | High | Urban—postnatal ward of a tertiary hospital in Basel | Longitudinal qualitative study utilizing an interpretive phenomenological approach | 15 mothers of various parities including participant observations in the postnatal ward and two narrative interviews at participants' homes at 6–8 and 12–14 weeks post-partum | A- |
| Maher JM, Souter K. *'It's much easier to get help for the baby': Women, postpartum health and maternal and child health care groups*. [38] | 2006 | Australia | High | Urban centres in Melbourne and regional centres in Victoria | Qualitative and descriptive informed by focus groups | 32 first time mothers in 4 focus groups conducted up to 12 months postpartum | C |
| Martin A, Horowitz C, Balbierz A, Howell EA. *Views of Women and Clinicians on Postpartum Preparation and Recovery*. [39] | 2014 | USA | High | Urban—teaching hospital in New York city | Qualitative and exploratory utilizing a 'Common Sense Model' and informed by focus groups | 45 mothers (mixture of parities) participating in 4 focus groups stratified by insurance type (Medicaid or private) interviewed 2–12 months postpartum | B+ |
| Martínez-Martínez A, Arnau J, Salmerón JA, Velandrino AP, Martínez ME. *The sexual function of women during puerperium: a qualitative study, Sexual and Relationship*. [40] | 2017 | Spain | High | Urban hospital in Murcia | Qualitative and exploratory using a phenomenological approach and informed by interviews | 30 women of various parities and stratified by various criteria interviewed 6–8 weeks postpartum | B |
| Mbekenga CK, Christensson K, Lugina HI, Olsson P. *Joy, struggle and support: Postpartum experiences of first-time mothers in a Tanzanian suburb*. [41] | 2011 | Tanzania | Low | Urban—2 clinics in a low-income suburb in Dar-es-Salam | Qualitative and descriptive informed by interviews | 10 first time mothers interviewed between 4 and 10 weeks postpartum | A- |
| Merighi MAB, Gonçalves R, Rodrigues IG. *Experiencing the postpartum period: A comprehensive approach of social phenomenology*. [42] | 2006 | Brazil | Upper Middle | Urban—private hospitals in a Brazilian city | Qualitative and exploratory using Alfred Schutz's theory of social phenomenology informed by interviews | 12 first time mothers in receipt of private health care interviewed 4–6 weeks postpartum | C+ |
| Nieterman E, Fox B. *Controlling the unruly maternal body: Losing and gaining control over the body during pregnancy and the postpartum period*. [43] | 2017 | Canada | High | Unclear. A range of settings in Ontario | Qualitative and descriptive informed by interviews | 63 women who were either pregnant or had given birth within the past two years interviewed by different authors at different time periods (42 in 2008–09; and 21 in 2012–13) | C+ |

*(Continued)*

**Table 1.** (Continued)

| Author(s) and Study | Year | Country | Resource | Context | Study design | Participants | Quality Rating |
|---|---|---|---|---|---|---|---|
| Pascual CP, Pinedo IA, Grandes G, Espinosa Cifuentes M, Gaminde Inda I, Payo Gordon J. *Perceived needs of women regarding maternity. Qualitative study to redesign maternal education.* [44] | 2016 | Spain | High | Unclear. Women residing in Bizkaia province | Qualitative and descriptive informed by focus groups | 31 primiparous women participating in 4 focus groups stratified by socioeconomic status and stage of the process (pregnancy vs. postnatal period) conducted within 4 months postpartum | A- |
| Price SL, Aston M, Monaghan J, Sim M, Murphy GT, Etowa J, et al. *Maternal Knowing and Social Networks: Understanding First-Time Mothers' Search for Information and Support Through Online and Offline Social Networks.* [45] | 2018 | Canada | High | Mixed—urban and rural locations across Nova Scotia | Qualitative and exploratory using feminist post-structuralism as the guiding philosophy & methodology and informed by focus groups, e-interviews and online forums | 19 first time mothers participating in 5 focus groups plus e-interviews with 18 first time mothers, all conducted within 1 year postpartum | A- |
| Razurel C, Bruchon-Schweitzer M, Dupanloup A, Irion O, Epiney M. *Stressful events, social support and coping strategies of primiparous women during the postpartum period: a qualitative study.* [46] | 2011 | Switzerland | High | Urban—Geneva hospital | Qualitative and descriptive informed by interviews | 60 first-time mothers interviewed 6 weeks postpartum | B+ |
| Riberio JP, Lima FBC de, Soares TMS, Bubolz Oliveira B, Voss Klemtz F, Barcelos Lopes K, et al. *Needs felt by women in the puerperal period.* [47] | 2019 | Brazil | Upper Middle | Unclear. A hospital in southern Brazil | Qualitative exploratory *and* descriptive study informed by interviews | 20 women of different parities including 10 that were interviewed in the immediate postpartum (up to 10 days after birth) and 10 in the remote postpartum (beyond 45 days) | C |
| Rotich E, Wolvaardt L. *A descriptive study of the health information needs of Kenyan women in the first 6 weeks postpartum.* [48] | 2017 | Kenya | Lower Middle | Urban—large referral hospital in a city (Eldoret) | Qualitative and exploratory informed by multiple interviews | 15 women of different parities interviewed at 3 time points:- within 48 hours postpartum, 2 weeks postpartum and 6 weeks postpartum | B |
| Runquist J. *Persevering Through Postpartum Fatigue.* [49] | 2007 | USA | High | Urban—a hospital in a city in Arizona | Qualitative and exploratory utilizing Grounded Theory methodology and informed by interviews | 13 women including 5 primiparous and 8 multiparous interviewed between 2 & 5 weeks postpartum (until data saturation achieved) | A |
| Salim NR, Junior HPOS, Gualda DMR. *Everyday behavioral and physical changes in women during the postpartum period—a qualitative approach.* [50] | 2010 | Brazil | Upper Middle | Urban—a maternity hospital in Sao Paolo | Qualitative and exploratory informed by observation and interviews | 6 first time mothers (aged 18–23) receiving care in a public hospital interviewed after 45 days postpartum | C |
| Salim NR, Araújo NM, Gualda DMR. *Body and Sexuality: Puerperas' Experiences.* [51] | 2010 | Brazil | Upper Middle | Urban—a maternity hospital in Sao Paolo | Qualitative and exploratory informed by observation and interviews | 6 first time mothers (aged 18–23) receiving care in a public hospital interviewed after 45 days postpartum | C |
| Salim NR, Gualda DMR. *Sexuality in the puerperium: the experience of a group of women.* [52] | 2010 | Brazil | Upper Middle | Urban—a maternity hospital in Sao Paolo | Qualitative and exploratory informed by observation and interviews | 6 first time mothers (aged 18–23) receiving care in a public hospital interviewed after 45 days postpartum | C+ |

*(Continued)*

**Table 1.** (Continued)

| Author(s) and Study | Year | Country | Resource | Context | Study design | Participants | Quality Rating |
|---|---|---|---|---|---|---|---|
| Slomian J, Emonts P, Vigneron L, Acconcia A, Glowacz F, Reginster JY, et al. *Identifying maternal needs following childbirth: A qualitative study among mothers, fathers and professionals.* [53] | 2017 | Belgium | High | Urban—2 large maternity hospitals in Liege | Qualitative and exploratory utilizing a multi-stage procedure and informed by focus groups and interviews | 22 women (mix of parities and ages) interviewed and grouped by:- 4–6 weeks postpartum with no PND symptoms (n = 10); 4–6 weeks postpartum with PND symptoms (n = 5); 10–14 months partum with no PND symptoms (n = 2) and 10–14 months partum with PND symptoms (n = 5). Also included 1 focus group with 5 women discussing partum needs | B- |
| Stevens J, Schmied V, Burns E, Dahlen H. *Skin-to-skin contact and what women want in the first hours after a caesarean section.* [54] | 2019 | Australia | High | Urban—large metropolitan hospital in Sydney | Qualitative and exploratory using video-ethnography informed by observation, field notes, focus groups and interviews | 21 women of various parities who underwent a c. section interviewed 6 weeks postpartum | A- |
| Tsai SY, Hu WY, Lee YL, Wu C-Y. *Infant sleep problems: A qualitative analysis of first-time mothers' coping experience.* [55] | 2014 | Taiwan | High | Urban—a medical centre in Northern Taiwan | Qualitative and exploratory using a transactional model of sleep/wake cycles informed by interviews | 15 first-time mothers interviewed at 3 months postpartum and recruited until data saturation achieved. | B+ |
| Way S. *A qualitative study exploring women's personal experiences of their perineum after childbirth: Expectations, reality and returning to normality.* [56] | 2012 | UK | High | Unclear—a hospital in SW England | Qualitative and exploratory utilizing Grounded Theory methodology and informed by interviews and diaries | 11 women of various parities asked to keep diaries for 10 days after birth and interviewed 2 weeks postpartum. Recruited until data saturation achieved | B+ |
| Wilkins C. *A qualitative study exploring the support needs of first-time mothers on their journey towards intuitive parenting.* [57] | 2006 | UK | High | Mixed—a regional hospital and 4 MW led units across the south of England | Qualitative and exploratory utilizing Grounded Theory methodology and informed by interviews | 8 primiparous women aged 20–39 years, who had given birth normally at term to a healthy baby were interviewed at 6 weeks postpartum. | A- |
| Yeh YC, St John W, Chuang YH, Huang Y-P. *The care needs of postpartum women taking their first time of doing the month: a qualitative study.* [58] | 2017 | Taiwan | High | Urban—a hospital in Taipei | Qualitative and descriptive informed by in-depth interviews | 27 first-time mothers interviewed in a Postpartum Nursing Centre ('doing the month') during their 35 day stay | C+ |
| Young E. *Maternal expectations: do they match experience?* [59] | 2008 | UK | High | Unclear—an area in South East England | Qualitative and descriptive informed by interviews and focus groups | 11 first time mothers interviewed up to a year postpartum. [Also includes focus group data from health professionals] | C |
| Zamawe CF, Masache GC, Dube AN. *The role of the parents' perception of the postpartum period and knowledge of maternal mortality in uptake of postnatal care: a qualitative exploration in Malawi.* [60] | 2015 | Malawi | Low | Unclear. An area in central Malawi with high levels of poverty and low uptake of postnatal care services | Qualitative and descriptive informed by interviews and focus groups | 36 first time mothers participated in focus groups up to 1 year postpartum. [Authors also conducted 1 focus group with 14 partners (fathers)] | B |

studies, studies based on grounded theory and ethnographic studies. They represent the views of more than 800 women, from a range of ethnic backgrounds and socio-demographic groups.

# Findings

We generated 22 descriptive themes (review findings), most of which were graded as high or moderate confidence. From these, we generated five analytical themes: *Riding the emotional rapids; Dancing around the baby–social and relational adaptation; It takes a community to raise a mother; Re-forming the birthing body*; and *Putting the mother into postnatal care*. Our framework and associated CERQual gradings are shown in Table 2.

From our analysis, we present the following summary statement;

*The postnatal phase is a period of significant transition characterised by changes in self-identity, the redefinition of relationships, opportunities for personal growth and alterations to sexual behaviour as women adjust to the 'new normal' in their own cultural context. Women describe experiencing intense joy and happiness, and also low mood, anxiety and depression, often compounded by feelings of acute fatigue and exhaustion. Women often prioritise their relationship with the new baby and sacrifice their own practical and self-care needs in favour of the baby's nutrition and safety. At the same time women may be struggling to come to terms with the physical and psychological impact of childbirth and the lack of opportunity to process associated feelings. This includes negative feelings associated with their post-birth body image. To cope with this period of adjustment women express the need for practical, emotional and psychological support from family members, peer groups and online sources, as well as from health providers. Women also want information and reassurance from health providers delivered in a consistent manner by authentic, familiar providers who recognise the mother's as well as baby's needs, within a well-resourced and flexible healthcare system that respects their cultural context.*

**Table 2. Framework of themes emerging from the data.**

| Descriptive theme (Review finding) | Studies contributing to the review finding | CERQual grading | Supporting data | Analytical theme |
|---|---|---|---|---|
| **Value feelings of overwhelming joy and happiness**–The birth of a baby can generate feelings of great happiness and joy and, for primiparous women in particular, unexpected sensations of overwhelming love (rarely or never experienced before). | **7 studies**–[26], [27], [33], [34], [41], [42], [47] | Low | *'Being a mother is wonderful so far. I think it will be really, really good. If I had known before, I would already have had a baby. It's been really good, I am very happy. It is so good! There is nothing negative about it.'* [34] | **Riding the emotional rapids** |
| **Coping with periods of low mood, depression and loneliness**–For some mothers the postnatal period is permeated with feelings of isolation and loneliness sometimes coupled with a sense of loss for their previous pre-natal self. Such emotions sometimes alternate with feelings of joy and love leaving them feeling conflicted. Some mothers also experience a loss of control and feelings of guilt for not being a "good mother". | **11 studies**–[25], [26], [28], [32], [33], [34], [37], [42], [46], [47], [53] | Moderate | *'One thing that is hard to understand, one thing you're not prepared for is the feeling of loneliness. I had this idea about us becoming a family, more than me becoming a mother, but then he had to work a lot, and the hours felt really long, and I felt very alone. I remember thinking, here I am, the baby is crying, it's dark outside, and there is actually no-one who really understand what I'm going through.'* [32] | |
| **Anxiety and fear about new responsibilities**–For some mothers, the responsibility of taking care of a baby is experienced as overwhelming or shocking. Mothers describe anxiety and worry about their baby's safety and wellbeing, and about aspects of baby care (e.g. feeding, whether to use dummies/pacifiers, how to soothe infant crying). Perceived social expectations to be a "good mother" and assumptions that other women are more competent mothers can generate feelings of anxiety. | **11 studies**–[30], [31], [34], [36], [37], [45], [46], [47], [53], [57], [59] | Moderate | *I wish there was a little switch. You suddenly realise there's no going back. It's not like a CD. If you don't like that you can take it back. It's now. This is for real. And that's the worst part of the first couple of weeks. It's like, 'Oh my God, I'm now responsible. I can't take it back. I can't drop it off somewhere.' It's just that realisation.* [57] | |
| **Coping with feelings of acute fatigue and exhaustion**–Women experience extreme tiredness and exhaustion, often compounded by a lack of sleep which can impact on many aspects of life. This can leave women feeling stressed and unable to focus or retain information (including information and advice given by health professionals). | **13 studies**–[26], [29], [34], [35], [37], [38], [40], [42], [44], [47], [49], [53], [55] | Moderate | *'It's not just physical tiredness. It's also emotional tiredness. It's the sleep that I am missing but it's also this new situation, all these impressive events . . . one has to adapt to this new situation and overcome the experience of the delivery. And one has to get to know the child.'* [35] | |

*(Continued)*

**Table 2.** (Continued)

| Descriptive theme (Review finding) | Studies contributing to the review finding | CERQual grading | Supporting data | Analytical theme |
|---|---|---|---|---|
| **Adjusting to new routines within local, cultural context**–Women describe challenges in establishing and/or adjusting to new routines and letting go of old ones. There may be competing demands e.g. between new routines centred around the baby and societal structures (baby-unfriendly society) or economic considerations (need to go back to work). Some women have cultural beliefs and practices pertaining to the postnatal period that they value and these may conflict with advice from healthcare providers. | **16 studies**—[26], [28], [33], [34], [36], [37], [40], [41], [45], [47], [48], [49], [52], [56], [57], [58] | High | "I was alone with my baby, in such a chaotic situation. I found myself wasting a lot of time. I just wandered around at home and couldn't do anything while all the house chores were left undone." [33] | **Dancing around the baby–social and relational adaptation** |
| **Prioritizing the needs of the baby**–Women tend to prioritize the needs of their baby above all else, sometimes at the expense of intimate concerns like their close relationships and their own self-care needs but also in relation to practical activities like domestic chores and household maintenance. Some experienced mothers were aware of this focus (after previous births) and actively decided not to put the baby's care needs above everything else. | **9 studies**—[35], [37], [38], [40], [42], [48], [49], [50], [58] | Low | 'I just realize that his needs are more important than mine and I'll find another time to sleep. It's just kind of an understanding of what's more important is probably the way I cope with it' [49] | |
| **Recognition of personal growth**–Women experience positive changes in their wellbeing ranging from an increased confidence and competence in their ability to mother to previously unacknowledged feelings of responsibility, calmness and perseverance, particularly amongst first-time mothers. Sometimes women are aware of an increase in their own sense of self-worth and express this in terms of being less selfish, more compassionate and more valued as a woman or as part of a family. | **11 studies**—[27], [32], [33], [34], [41], [42], [47], [49], [50], [55], [57] | Moderate | 'So I am a person who has changed a lot. I was stressed out and jumpy but I became a lot calmer after I had her. She has calmed me down.' [50] | |
| **Challenges of breastfeeding**–Women value breastfeeding and want support to be available. However, they often find the practical reality of breastfeeding at odds with their expectations and experience anxiety, frustration and sometimes pain in their attempts to establish the practice. | **8 studies**–[30], [34], [36], [38], [44], [46], [48], [54] | Moderate | 'With the caesarean section, the milk was cut off [. . .], I had a terrible time not breastfeeding because I wanted to breastfeed, then you realize that well, it has come like this [. . .]. But at that moment [. . .] the world comes to you [. . .] and you say "why cannot I breastfeed?"' [44] | |
| **Dealing with changes in sexual behaviour**–Women have to deal with a variety of different sex related issues in the postnatal period and would like more information from health professionals, particularly about when to resume sex and access contraceptive resources. Women may experience changes in their sex drive (positive or negative), painful sex, unsatisfactory sex or feel reluctant to have sex at all in case they become pregnant again. Some women feel embarrassed or ashamed about having sex in the same room as the baby, feel anxious about satisfying the needs of their partner or feel too exhausted for sex. | **8 studies**–[26], [29], [40], [41], [48], [50], [51], [52] | Low | 'When it [finally] happened, yeah [I had interest]. But not so much in the beginning. Like, no that's the last thing . . . Everything is closed for business. Off limits. I gotta heal man'. [29] | |
| **Redefining identity and relationships**–Some women find the transition to motherhood difficult and feel insecure with the new identity. They struggle with the loss of the 'familiar self' and their desire to spend some time alone time and this can cause tension in their relationships with partners or parents | **9 studies**—[26], [27], [33], [41], [43], [51], [56], [58], [59] | Moderate | ''He thinks that you are not listening to him; he says 'when I talk you don't listen. Since you had that child, you have been ignoring me, it's only about that child, nothing else.' I cannot help him; I don't know how I can do that! There is really no way to help him because most of the time I am with my child' [41] | |
| **Developing a relationship with the baby**–Women want to form an immediate relationship with their baby and try to achieve this by lots of contact and stimulation (e.g. gazing, hugging, kissing and singing) sometimes finding it difficult to leave the baby alone. Recognizing, understanding and responding to different crying cues is particularly important for first time mothers as a way of facilitating this relationship. Women who deliver their baby via caesarean section appreciate immediate skin to skin contact as a way of initiating a relationship with their baby. | **9 studies**—[34], [35], [36], [37], [47], [48], [54], [55], [58] | Moderate | 'I readily realize when she just wants to be occupied or doesn't want to be alone, when she is bored or feels really bored, when she doesn't want to be by herself, or if she has really had enough. I realize if she just wants some occupation, or when she is frustrated because something does not go her way. Just, when she is hungry, when she cannot sleep or when she just wants to cry.' [36] | |

*(Continued)*

**Table 2.** (*Continued*)

| Descriptive theme (Review finding) | Studies contributing to the review finding | CERQual grading | Supporting data | Analytical theme |
|---|---|---|---|---|
| **Importance of practical and emotional support from partner, family and elders**–Most women greatly value the multiple levels of support they receive from their partner, parents and, in certain contexts, community elders. Women appreciate practical help with the baby's care needs and household tasks as well as emotional support to help them deal with the myriad of emotions that arise during the postnatal period. Some women occasionally report negative experiences (e.g. lack of breastfeeding support, interference or undermining behaviour). | **18 studies**–[26], [27], [28], [30], [31], [32], [33], [36], [37], [38], [41], [47], [49], [53], [54], [55], [58], [59] | High | 'My mum knows what my favourite foods are and what I need for doing the month. She cooks meals at home and brings them to me. She reminds me to pay more attention to food now because this is vital for women who have given birth. [58] | **It takes a community to raise a mother** |
| **Value practical advice, information and support from health professionals**–Women value support, reassurance and information from health professionals especially around baby development, hygiene, vaccinations, breastfeeding, nutrition, and practical caregiving tips. Some women would like to be given more information in the early postnatal phase (prior to discharge), whilst others would prefer to be given information during the antenatal phase. | **16 studies**–[26], [27], [30], [31], [34], [35], [37], [39], [41], [44], [46], [48], [49], [53], [54], [57] | High | "I think the nurses were really helpful, taking the time to stay with you and showing you how, or trying to find different techniques" [31] | |
| **Value, information, support and reassurance from peers and peer groups**–Women value the emotional support, reassurance and informational resources they acquire from meeting with other mothers (especially first-time mothers) in formal or informal peer support groups. They appreciate the non-hierarchical nature of these groups and the normalization of perceived insecurities about being a "good mother". In some contexts, women want peer support but are unable to attend regular meetings (costs) whilst others advocate a "peer support DVD" in which first-time parents would recount their experiences in an informal, authentic manner. | **15 studies**–[26], [27], [28], [32], [33], [38], [41], [43], [44], [45], [46], [52], [53], [57], [59] | High | 'Yeah, we all get together and sort of lash things out and talk, babies . . .and you know everybody learns from their own experiences and if we all can sort of contribute, that's really good.' [27] | |
| **Importance of online sources of support and information**–Some women (especially first-time mums) find benefit in online support groups and web-based sources of information and some wanted an online guide to address common postnatal concerns. However, others found too much online searching added to their levels of stress and anxiety. | **4 studies**–[38], [44], [45], [53] | Low | 'Reading and looking for advice online was good in that it was on my own time, often in the middle of night while nursing and I could take what I wanted and leave what I didn't in terms of advice, I wasn't going to offend anyone'. [45] | |
| **Coping with labour and birth induced trauma**–Women feel unprepared for the physical and psychological effects of labour & birth induced trauma and the impact this has on their ability to provide appropriate care for their baby (and other children). Women experience feelings of fear and anxiety associated with the long-term management of caesarean section wounds, perineal damage, bladder problems, vaginal bleeding and general discomfort. Some women would like more information from health professionals about how to soothe/treat physical injuries and some would welcome the opportunity to discuss their labour and birth with a healthcare provider. | **15 studies**–[28], [31], [32], [38], [39], [40], [43], [47], [48], [49], [50], [52], [56], [59], [60] | High | 'Was this expected? Bizarrely no! Thinking about it now I cannot understand what I thought it would feel like. Obviously I was under no misapprehension that actually giving birth would be painful although how much I had no idea. I had never given it any thought to how I would feel afterwards.' [56] | **Re-forming the birthing body** |
| **Coming to terms with changes in body image**–Some women struggle with feelings of shame, embarrassment and insecurity about their post-birth body image. Unrealistic expectations of how quickly they can get 'back into shape' sometimes leads to frustration and changes in their sexual identity, e.g. viewing their breasts in a non-sexual way or experiencing a reduction in their sex drive, may also cause concerns. | **6 studies**—[25], [29], [40], [50], [52], [56] | Low | 'Now I look at my stomach and think 'well, now she's not in there it's not so good' . . . you have a baby and then you're left with a big stomach and then you've got to try to get rid of it.' [25] | |

**Table 2.** (Continued)

| Descriptive theme (Review finding) | Studies contributing to the review finding | CERQual grading | Supporting data | Analytical theme |
|---|---|---|---|---|
| **Value continuity of care**–Women want to build a relationship with health professionals during the postnatal period and may become frustrated if continuity of care isn't available | **5 studies**–[27], [30], [37], [39], [42] | Low | '. . .in those two weeks you've built up such a relationship. I felt comfortable and . . . a few weeks is not long enough for breastfeeding and all that . . . When she left I felt very alone. . . . Because she takes all my notes and I didn't have any contact numbers for the health visitor. What do I do in the next week?. . . . A relationship with someone, it's trust isn't it?' [27] | **Putting the mother into postnatal care** |
| **Need for consistent information**–Women want consistent information from health professionals (e.g. around breastfeeding, care practices, nutrition and clinical or informal observations) and express frustration when information is conflicting or inconsistent | **7 studies**–[26], [30], [37], [45], [46], [48], [54] | Moderate | 'one person will tell you give your baby a certain type of food. . .another one will tell you not that one. . .another one will tell you apply something to your baby. . .now all this information. . .which one do you follow?' [48] | |
| **Value woman-centred care**–Women want to feel 'cared for' during the immediate postnatal period and would like personal preferences to be acknowledged by health professionals. These may include not feeling pressure to breastfeed, recognizing and acting on maternal intuition, involving partners in caregiving practices and being more flexible in meeting women's needs. More tailored support for mothers of pre-term babies and more information on how to cope after a caesarean section are also important considerations for some women. | **10 studies**–[26], [27], [30], [32], [35], [37], [39], [45], [48], [58] | Moderate | "I was always so grateful when she was ringing the bell. Even if she was just observing and then telling me how good my child and I were doing. Or giving me some advice or helping me understand my child who did cry a lot. This was so important to me and I still benefit from this now." [37] | |
| **Need for greater emphasis on mothers' wellbeing**–Although women recognize and appreciate the attention on the postnatal care needs of their baby they sometimes feel their own emotional and psychosocial needs are overlooked or neglected by health professionals. | **10 studies**–[28], [31], [32], [38], [39], [42], [46], [53], [54], [60] | High | 'When I went to see [the doctor] at the six weeks check-up, it seemed to be almost more about her than me!' [38] | |
| **Organizational systems hinder effective care**–Women may experience poor or inadequate care in the immediate post-partum period which may be exacerbated by organizational limitations (shared rooms, lack of privacy, poor hospital condition) and disruptive hospital routines (ward rounds, visiting hours). Perceived staff shortages leads to delays in calls for support and assistance and some women feel they are discharged from hospital too early. Women value postnatal home visits but occasionally feel they don't receive enough and find health professionals difficult to access when they are needed. | **9 studies**–[26], [30], [31], [35], [38], [39], [41], [47], [60] | Moderate | 'I was uncomfortable, I didn't sleep, the girl next me had a baby that was crying all night.' [30] | |

Based on this statement we identified the key components of a positive postnatal experience to align with previous reviews of what matters to women during antenatal care [61] and intraprtum care [62] (see Box 2)

We now discuss each of the five analytical themes in more detail below.

## Riding the emotional rapids

Women experience a range of conflicting and contrasting emotions during the postpartum period, from intense feelings of joy and love for their new baby, to acute feelings of loneliness,

> ### Box 2. Positive postnatal experience.
>
> A positive postnatal experience is one in which women are able to adapt to their new self-identity and develop a sense of confidence and competence as a mother, adjust to changes in their intimate and family relationships, including their relationship to their baby, navigate ordinary physical and emotional challenges, and experience the dynamic achievement of personal growth as they adjust to the 'new normal' of motherhood and parenting in their own cultural context.

low mood and depression. Women may experience guilt for not living up to pre-conceived notions of 'the ideal mother' and feel burdened by the weight of responsibility involved in caring for their baby. This can lead to feelings of being out of control and, for first-time mothers especially, a lack of confidence in their ability to soothe, provide for and care for the baby. These emotions are often intensified or triggered by overwhelming feelings of fatigue as women come to terms with the practicalities of caring for a new baby and the associated changes in sleep patterns. Women may feel exhausted and exasperated and unable to process or retain information, especially during the early postnatal phase, when they are trying to adapt to major lifestyle changes.

### Dancing around the baby–social and relational adaptation

Emotional and psychological adjustments may be compounded by social and relational adjustments and abrupt changes of old routines to new ones. Women prioritize the needs of their baby above all else and struggle to accommodate previously simple practical tasks like household chores, self-care and leaving the house. Women want to form an immediate bond with their baby, especially after a caesarean section, and want to cope with the challenges of breastfeeding and recognize different crying cues to develop this bond. The emphasis on the baby can place a strain on intimate relationships as women juggle with caring for the baby, finding time to be alone and maintaining a relationship with their partner. Some relationships may become strained as couples adjust to the new family unit and this can impinge upon sexual activity. Knowing when to resume sexual relations is a cause of concern for some women and alterations in libido may create additional anxieties. For other women the postnatal period brings socio-cultural expectations, including the tradition of 'doing the month', and, although this is largely welcomed by new mothers, it can highlight family tensions and be a source of misunderstanding by healthcare professionals. As women navigate the various challenges of the postnatal period changes in self-identity emerge and they begin to value the sense of being recognized as 'a woman' and 'a mother'. Women also begin to appreciate their previously unacknowledged capacity to persevere, to love, to be selfless, to be compassionate and, ultimately, to feel more complete.

### It takes a community to raise a mother

Women value the practical and emotional support they receive from their partners, family and (in some contexts) community elders. Sometimes women report negative experiences such as unsupportive, interfering or undermining behaviour from family members, but more often women receive emotional support, and practical help with household tasks and caring for the baby. Healthcare providers can also have a role in offering reassurance, validation and guidance: women want information on baby development, hygiene, vaccinations, breastfeeding,

nutrition, and practical caregiving tips, with some preferring to receive this antenatally and some in the early postnatal period. Women also value support from other mothers, for example via peer support groups, to share experiences, ease insecurities and share information. This was particularly pertinent in studies of first-time mothers. Some women find online support groups and information helpful, although others describe finding too much use of online material to be anxiety provoking.

## Re-forming the birthing body

Women feel unprepared for the physical and psychological effects of labour and birth and the course of recovery in the postnatal period. Women feel anxiety about injuries such as caesarean section wounds, perineal damage, bladder problems, vaginal bleeding and general discomfort, and some women would like more information from health professionals about how to soothe or treat physical injuries. Some women feel the need to discuss their labour and birth with a healthcare provider. During the postnatal period, women adjust to how pregnancy and birth have altered their bodies and some struggle with feelings of shame, embarrassment and insecurity. Some women hold beliefs (related to perceived social pressure) about needing to lose weight and/or regain fitness. Women's sexual desire may be affected by insecurities about their postnatal bodies, or by changes in their perceptions of their bodies (such as breasts now perceived as non-sexual).

## Putting the mother into postnatal care

Women want to feel 'cared for' during the postnatal period as they navigate the transition to motherhood and recovery from labour and birth. They want to be seen as individuals and for their needs as well as their baby's needs to be recognised and met. Good quality care in the postnatal period includes continuity of care provider to enable a trusting relationship, consistent information, and flexibility and recognition of women's personal and cultural contexts. Women may struggle with organisational factors such as shared rooms and lack of privacy, and disruptive hospital routines such as ward rounds and visiting hours. Women may feel they have had inadequate support due to being discharged home early, receiving too few postnatal visits, or not receiving timely support.

## Discussion

Based on a systematic qualitative review of findings from 36 studies from 15 countries published between 2003 and 2019 [25–60], the postnatal phase is a period of significant transition characterised by changes in self-identity, the redefinition of relationships, opportunities for personal growth, and alterations to sexual behaviour as women adjust to the 'new normal' within their own cultural context.

As Rubin has noted, becoming a mother, especially for the first time, involves '*an exchange of a known self in a known world for an unknown self in an unknown world*' [63], p52. As Lanley et al comment, for this dynamic adaptive process to be successful, previous identities need to be remade [64], '*the process of incorporating motherhood into women's identities can be conceptualized as a fracturing of identity wherein women lose or have compressed selves for a time. This fracturing creates space for women to incorporate their children's needs into their awareness*'. In their phenomenological study of Australian women in the postnatal period, Rogan et al [65] cite Rubin, and describe the fluid nature of the adaptation to becoming a woman who is also a mother. While this shift is dramatic when it occurs for the first time, subsequent childbirth requires a new process of re-calibration as the identities of 'woman' and 'mother' once more shift. In line with Rogan, and with our prior expectations, our review notes the difficulties and

challenges of this process. However, unlike the mostly negative accounts of Rogan's cohort of Australian women [63], inclusion of studies from a wide range of social and cultural contexts in our review has also illuminated the life-enhancing, joyful aspects of the early postnatal period.

Based on our findings, for this transition to motherhood to be a positive experience, women need to balance both losses and gains. Losses encompass changed (often negative) body image, reduced capacity to be in control of one's time and sleep, changes to previous sexual and romantic relationships with partners, and a loss of identity as an individual with inherent value (separate from the baby). Gains include intense feelings of joy and love for their new baby, and a new sense of triumph and self-esteem in overcoming the difficulties and stresses of pregnancy, birth, and the postnatal period, as well as the discovery of a previously unanticipated capacity to persevere, to love, to be selfless, to be compassionate and, for many, to feel more complete in their new integrated identity of 'woman as mother'.

To enable optimum adaptation and wellbeing, women's accounts suggest that they value practical, emotional and psychological support from family members, peer groups and online sources. Our findings here are in accord with a range of studies from different contexts and cultural settings where social support is seen as a vital component in effective transition to motherhood [66–68]. In adapting the often-cited phrase *'it takes a village to raise a child,'* into *'it takes a community to raise a mother'* we have tried to capture the essence of the data that showed that the primary source of support and growth in this period is the family and the local community in which new mothers are embedded. This suggests that, to ensure that all women experience a positive postnatal experience, families and communities should be informed, educated and mobilised so that women receive realistic information, appropriate reassurance, and culturally appropriate validation. Where women do not have access to supportive families and communities, or where there is conflict between the woman and her family/community, or, more generally, with the norms and expectations of being mother and of childcare that are held by her family/community, her capacity to cope and to successfully integrate the identities of both woman and mother could be more difficult and problematic. This may be a time when good quality professional input is especially valuable and effective.

In our previous review of what matters to women at the beginning of the maternity episode, in the antenatal period, adaptation to motherhood was seen as the critical endpoint of the pregnancy and labour experience. This was the case even though the actual nature of motherhood wasn't clear, especially to those who were having their first baby [61]. This current review of women's accounts at the end of the postnatal period reinforces the sense that, for childbearing women around the world, maternity care is not perceived as three distinct stages (antenatal, intrapartum, postnatal) as defined by most maternity care systems, but as an adaptive process that dynamically and continuously builds and reframes women physically, psychologically, socially and emotionally through birth and motherhood.

It is unsurprising that the hallmarks of good quality professional postnatal care are the same as those for good antenatal and intrapartum care–support, communication, and effective clinical input, based on what each mother and baby and family need, when they need it, and how they need it [61, 62]. This finding is so prevalent across studies of pregnancy, childbirth, and the postnatal period, in all contexts and cultures, that there could be an argument that health systems that do not provide this kind of integrated maternity care are malfunctioning. These components are especially important for women who do not have community and family support for whatever reason.

Whilst we didn't seek to explore the specific requirements of first-time mothers our review does include nineteen studies of this specific population and our findings indicate that the theme related to support and information ('it takes a community to raise a mother') is likely to

be particularly relevant to this group, regardless of setting or context. Our findings represent the views of more than 800 women from a wide variety of settings and contexts including Europe, Africa, North and South America, Australasia and the Middle East. As such, they bring together the perspectives of numerous, disparate, individuals and, given the relatively high CERQual gradings (16/22 were graded high or moderate), we can be reasonably confident they accurately portray shared understandings relating to what matters to women during the postnatal period. Although this is a major strength, we could only find three studies from LICs. This is a limitation, as there are likely to be particular cultural insights relating to postnatal care practices in these contexts. Most of our included studies were done in urban settings and we cannot rule out that women in rural settings may have different experiences. In addition, our searches were carried out in health and medical databases and it is possible that the inclusion of social science databases may have identified more papers relating to women's needs during the postnatal period.

Although our aim was to understand what women value during the postnatal period as opposed to their experiences of current provision, participant's values were occasionally conflated with their perceptions of the care they received. We therefore have some findings that relate to what matters about formal postnatal care provision, as well as about what women value in relation to experiencing the state of being postnatal. We were also unable to disentangle women's views and priorities relating to defined postnatal stages, i.e. early, intermediate and late, since most study authors didn't ask women to consider these specific time periods. Further qualitative research with women during these distinct phases may generate additional insights relating to the needs and priorities of mothers at different stages of the postnatal period.

Finally, we recognize that our review focuses on the values of healthy women from the general population and does not incorporate the views of women with identified clinical complications (e.g. cardiovascular disease, infection, diabetes, urinary problems) or mental health concerns (e.g. postnatal depression) that may require specific postnatal interventions.

## Conclusion

Our systematic scoping review of findings from 36 studies from 15 countries published between 2003 and 2019 demonstrates that a positive postnatal experience is one in which women are able to adapt to their new self-identity and develop a sense of confidence and competence as mothers, adjust to changes in their intimate and family relationships, including their relationship to their baby, navigate ordinary physical and emotional challenges, and experience the dynamic achievement of personal growth as they adjust to the 'new normal' of motherhood and parenting in their own cultural context. Effective, culturally appropriate family, community and professional support and activities can help women to overcome the exhaustion, and physical, emotional and psychological stress of the early postnatal period. As anticipated by a previous review of what matters to women in pregnancy, what matters to women in the postnatal period is achieving positive motherhood (including maternal self-esteem, competence, and autonomy). This review provides evidence that, where this process is positive, it also results in joy, self-confidence, and an enhanced capacity to persevere and to succeed in the new integrated identity of 'woman and mother'.

## Supporting information

**S1 Table. PRISMA checklist.**
(DOCX)

**S1 Appendix. Data extraction, analysis, synthesis and CERQual grading.**
(XLSX)

## Acknowledgments

We would like to thank all members of the WHO technical working group for their support and advice during the development of this review. This paper reflects the views of the named authors only, and not the views of their institutions.

## Author Contributions

**Conceptualization:** Mercedes Bonet.

**Data curation:** Kenneth Finlayson, Nicola Crossland.

**Formal analysis:** Kenneth Finlayson, Nicola Crossland.

**Funding acquisition:** Mercedes Bonet.

**Investigation:** Mercedes Bonet.

**Methodology:** Kenneth Finlayson, Mercedes Bonet, Soo Downe.

**Project administration:** Kenneth Finlayson, Nicola Crossland.

**Resources:** Soo Downe.

**Supervision:** Mercedes Bonet, Soo Downe.

**Writing – original draft:** Kenneth Finlayson, Nicola Crossland.

**Writing – review & editing:** Kenneth Finlayson, Nicola Crossland, Mercedes Bonet, Soo Downe.

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
