## [Decision Letter · Decision Letter 0]

21 Feb 2020

PONE-D-20-01368

What matters to women in the postnatal period: A qualitative systematic review

PLOS ONE

Dear Mr Finlayson,

Thank you for submitting your manuscript to PLOS ONE. After careful consideration, we feel that it has merit but does not fully meet PLOS ONE’s publication criteria as it currently stands. Therefore, we invite you to submit a revised version of the manuscript that addresses the points raised during the review process.

This qualitative systematic review of what mattes to women in the postnatal period is masterfully conducted and reported.

One reviewer has suggested the use of the eMERGe framework for reporting (https://bmcmedresmethodol.biomedcentral.com/articles/10.1186/s12874-018-0600-0#Sec9). I note that the previous reviews conducted by your group include antenatal and intrapartum themes and were conducted and reported using the PRISMA checklist. That justifies the use of PRISMA in this manuscript to quite an extent. In looking eMERGe, however, it would seem relatively simple to also note how it has been followed, given that nearly all the elements are included in your review – this would enhance how the review has also accounted for emergent thinking. In particular, you could address element 13 or eMERGe, giving some comment on how translations using Google Translate adequately represent the intended meaning and/or nuances of the original language.  Please also consider including a Summary of Findings table as an Appendix / Supplemental material, as this will give more detail about the CERQual scores ratings assigned.

We would appreciate receiving your revised manuscript by Apr 06 2020 11:59PM. To enhance the reproducibility of your results, we recommend that if applicable you deposit your laboratory protocols in protocols.io, where a protocol can be assigned its own identifier (DOI) such that it can be cited independently in the future. For instructions see: http://journals.plos.org/plosone/s/submission-guidelines#loc-laboratory-protocols

We look forward to receiving your revised manuscript.

Kind regards,

Christine E East

Academic Editor

PLOS ONE

Journal Requirements:

2. Thank you for stating the following in your manuscript:

"The work was commissioned to the University of Central Lancashire by the UNDP/UNFPA/UNICEF/WHO/World Bank Special Programme of Research, Development and Research Training in Human Reproduction (HRP), a cosponsored program executed by the World Health Organization (WHO)"

Reviewers' comments:

Reviewer's Responses to Questions

**Comments to the Author**

1. Is the manuscript technically sound, and do the data support the conclusions?

Reviewer #1: Yes

Reviewer #2: Yes

2. Has the statistical analysis been performed appropriately and rigorously? 

Reviewer #1: Yes

Reviewer #2: N/A

3. Have the authors made all data underlying the findings in their manuscript fully available?

Reviewer #1: Yes

Reviewer #2: Yes

4. Is the manuscript presented in an intelligible fashion and written in standard English?

Reviewer #1: Yes

Reviewer #2: Yes

5. Review Comments to the Author

Reviewer #1: General comment

The study provides new knowledge on an important aspect of maternity care that has received minimal attention previously contributing to the negative outcome to both the mother and baby.

Specific comments

As alluded to, the finding needs to be included in the guidelines.

Abstract line 24 indicates that postnatal care is undervalued- I expected this to be elaborated more in the introduction section more so in the first paragraph. If not possible consider using another term instead of undervalued. You may consider the term underserved as used in line 134

The aim in abstract need to be aligned with the aim in line 110 to 112.

Add reference to this statement in line 71-For some mothers and babies it includes ill-health.

Line 25 and 26 refers to a WHO guideline which is not cited in the introduction section

Line 74, - revise the sentence.

Line 98, - do you need the punctuation?

Line 98- ensure consistence of the word postnatal

Line 102- add “s “to LMIC

Line 104- remove the bracket after the reference

Line 178-180 add the reference

Revisit the two statement in line 201 and in figure 1- “excluded by” seems inappropriate,

Line 205 -indicate the actual number and avoid using “almost”, from the table, around 12 studies scored C

Line 385-include LMICs

Line 403-mothers instead of mother

A number of studies included were in urban set up, does it have any implication.

Reviewer #2: Thank you for asking me to review this qualitative evidence synthesis on what matters to women in the postnatal period. This is a well written paper producing novel conceptual findings through a synthesis of the findings from 36 qualitative studies. These synthesised findings bring together perspectives from women in a range of settings globally, and are organised as five overarching conceptual themes to capture women’s experience of the postnatal period.

Approach: I wonder, however, if “meta-ethnography” is an appropriate description of what has been done here, since much of it reads more like a thematic synthesis. p.9 line 183 refers to first order constructs being identified in the source material, and second order constructs being developed from these. This doesn’t map to my understanding of this terminology or how meta-ethnography uses these. First order constructs are everyday interpretations of the world (seen in participant quotes, and descriptive researcher findings), while 2nd order constructs are researchers’ interpretations of these 1st order constructs, including links to the wider theoretical literature. Meta-ethnography works with these more interpretative 2nd order constructs to provide further interpretations through the synthesis and usually leads to developing explanatory theory, or a conceptual model. These reviewer interpretations are sometimes referred to as third order constructs. It is also worth noting that Noblit and Hare consider meta-ethnography difficult for larger numbers of studies/ findings, which this synthesis arguably contains.

Typically, this approach requires well developed existing conceptualisation in the papers identified and, where these are not available, other synthesis approaches (framework, thematic for example) may be more appropriate. This may be what has happened here. Another explanation is that these stages have been undertaken but not all have been reported. The authors use adapted PRISMA reporting standards but it would have been better to use the eMERGe guidance which is designed specifically for reporting meta-ethnography. It may be that using eMERGe would clarify the extent to which the meta-ethnographic process has been possible (https://bmcmedresmethodol.biomedcentral.com/articles/10.1186/s12874-018-0600-0#Sec9). Key aspects of reporting a meta-ethnography across the seven phases outlined in France et al are currently missing.

I like the consolidated “positive experience” in box 2, but this isn’t referred to in the text, and some explanation in the methods of how this was developed would be helpful. Is there a commensurate “negative experience” exemplar?

I was interested to see that nearly all the findings were rated as moderate or high confidence using CERQual, although the review question appears to be global, and only three of the studies relate to LMIC. I might therefore expect greater downgrading of “relevance” in the CERQual scores for these findings which don’t contain geographical limits. Typically, the CERQual scores are accompanied by a Summary of Findings table, which includes a brief explanation about how the overarching CERQual score was derived. This might improve the transparency of how the gradings were assigned (see table 4 https://journals.plos.org/plosmedicine/article?id=10.1371/journal.pmed.1001895).

Searches: No social science data base (such as ASSIA) was searched – is it possible that papers have been missed? Can a reference be provided for the qualitative research filter used in the searches on p.7? I am unclear of the rationale for requiring line 5 AND line 6.

Minor points: It would be good to see data analysis method, as well as collection method, recorded in Table 1.

I’d be interested in the authors’ reflections on the appropriateness of using google translate in maintaining conceptual and idiomatic meaning for qualitative analysis, although this may be outside the paper’s scope!

There are some inconsistencies of language to iron out – for example, this paper is referred to as a “systematic scoping review” on line 401, which I think undersells the synthesis. I also suggest that the language of “overarching themes” rather than “summary themes” is kept throughout as the latter doesn’t capture the conceptual novelty of these themes.

6. PLOS authors have the option to publish the peer review history of their article (what does this mean?). If published, this will include your full peer review and any attached files.

Reviewer #1: No

Reviewer #2: No

---

## [Author Response · Author response to Decision Letter 0]

18 Mar 2020

Reviewers’ comments and Authors’ responses

Reviewer 1 

Abstract line 24 indicates that postnatal care is undervalued- I expected this to be elaborated more in the introduction section more so in the first paragraph. If not possible consider using another term instead of undervalued. You may consider the term underserved as used in line 134

 As suggested, we have amended the abstract to use the term underserved instead of undervalued. In addition, the Introduction (lines 96-108) includes the statement that postnatal care is a neglected area of maternity care, and outlines issues of emphasis, coverage, and utilisation. We trust this addresses the reviewer’s concerns.

The aim in abstract need to be aligned with the aim in line 110 to 112.

 We have amended the abstract to align with the aim in the Introduction, while keeping the abstract concise.

Add reference to this statement in line 71-For some mothers and babies it includes ill-health.

 We have added references for this statement.

Line 25 and 26 refers to a WHO guideline which is not cited in the introduction section

 This guideline is currently under development and has not yet been published; the current review was conducted in order to inform the guideline development.

Line 74, - revise the sentence.

 We are not sure what needs to be revised in this sentence. Some further clarification would be welcome.

Line 98, - do you need the punctuation?

 Thank you, we have removed the unnecessary comma from Line 98.

Line 98- ensure consistence of the word postnatal

 We have amended the manuscript to ensure ‘postnatal’ is consistent throughout.

Line 102- add “s “to LMIC

 In Line 102 we refer to ‘LMIC settings’ so it is appropriately written without the ‘s’ in this instance.

Line 104- remove the bracket after the reference

 We have removed the bracket.

Line 178-180 add the reference

 These lines describe the procedure we followed in conducted the meta-synthesis; the reference is given where the procedure is introduced in the previous sentence.

Revisit the two statement in line 201 and in figure 1- “excluded by” seems inappropriate,

 We believe ‘excluded by’ is an appropriate description of how records were selected based on our inclusion and exclusion criteria.

Line 205 -indicate the actual number and avoid using “almost”, from the table, around 12 studies scored C

 As suggested, we have amended this section to give the numbers of studies rated A, B and C on quality appraisal.

Line 385-include LMICs

 Here we are specifically referring to low-income countries, not low and middle-income countries, hence LIC.

Line 403-mothers instead of mother

 We have amended as suggested.

A number of studies included were in urban set up, does it have any implication.

 We have added to the Discussion section (Line 389) to note the most of the studies in the review were conducted in urban settings.

Reviewer #2

Approach: I wonder, however, if “meta-ethnography” is an appropriate description of what has been done here, since much of it reads more like a thematic synthesis. p.9 line 183 refers to first order constructs being identified in the source material, and second order constructs being developed from these. This doesn’t map to my understanding of this terminology or how meta-ethnography uses these. First order constructs are everyday interpretations of the world (seen in participant quotes, and descriptive researcher findings), while 2nd order constructs are researchers’ interpretations of these 1st order constructs, including links to the wider theoretical literature. Meta-ethnography works with these more interpretative 2nd order constructs to provide further interpretations through the synthesis and usually leads to developing explanatory theory, or a conceptual model. These reviewer interpretations are sometimes referred to as third order constructs. It is also worth noting that Noblit and Hare consider meta-ethnography difficult for larger numbers of studies/ findings, which this synthesis arguably contains. 

 Thank you for these insightful comments. As a team we are philosophically aligned with meta-ethnography and have tended to use this methodology in our previous syntheses. We also appreciate that ‘meta-ethnography’ is a distinct and often over-used descriptor for the synthesis of qualitative data and are aware of ongoing debates about the misappropriation of the term. We are also aware of the eMERGe guidelines and the focus on retaining the integrity of the meta-ethnographic method. In this instance we tried to ‘adapt’ some of the principles of meta-ethnography to suit a synthesis of qualitative findings within the context of a larger exercise – to inform the WHO guidelines on postnatal care. Unfortunately WHO Evidence to Decision (EtD) frameworks lend themselves more to the development of descriptive findings rather than conceptual or theoretical explication and, on reflection, we have probably compromised too far in our endeavours to adapt meta-ethnography to suit these frameworks. Upon further reflection we agree with the reviewers observation that our methodological approach is more aligned with a thematic synthesis and, having checked Thomas & Harden’s (2008)* description of the approach, feel that it should be labelled as such. We note that there are synergies between the meta-ethnographic approach and thematic synthesis as highlighted by Thomas & Harden (2008). We have compared our methods of appraisal, coding of data, development of themes and synthesis with those of Thomas & Harden (2008) and have found broad agreement. With this in mind we have relabelled our 2nd order constructs as ‘descriptive themes’ and our overarching 3rd order constructs as ‘analytical themes’ in accord with the thematic synthesis method. In addition, we note that thematic synthesis requires quality appraisal of included papers (which we have done) and is also more suitable for larger numbers of studies (Booth et al, 2016)**. We have removed references to Noblit & Hare (1988) to aid clarity.

* Thomas J, Harden A. Methods for the thematic synthesis of qualitative research in systematic reviews. BMC Med Res Methodol 8, 45 (2008). https://doi.org/10.1186/1471-2288-8-45.

** Booth, A, Noyes J, Fleming K, et al. (2016) Guidance on choosing qualitative evidence synthesis methods for use in health technology assessments of complex interventions [Online]. Available from: http://www.integrate-hta.eu/downloads/

Typically, this approach requires well developed existing conceptualisation in the papers identified and, where these are not available, other synthesis approaches (framework, thematic for example) may be more appropriate. This may be what has happened here. Another explanation is that these stages have been undertaken but not all have been reported. The authors use adapted PRISMA reporting standards but it would have been better to use the eMERGe guidance which is designed specifically for reporting meta-ethnography. It may be that using eMERGe would clarify the extent to which the meta-ethnographic process has been possible (https://bmcmedresmethodol.biomedcentral.com/articles/10.1186/s12874-018-0600-0#Sec9). Key aspects of reporting a meta-ethnography across the seven phases outlined in France et al are currently missing.

 Again, we appreciate the reviewers clarification here and hope that the reframed methodological narrative better reflects the methods employed. We haven’t referred to the eMERGe guidelines as they are no longer appropriate. 

I like the consolidated “positive experience” in box 2, but this isn’t referred to in the text, and some explanation in the methods of how this was developed would be helpful. Is there a commensurate “negative experience” exemplar?

 Thank you for bringing this to our attention. By way of explanation our research question was framed to capture “what matters to women in the postnatal period” and, whilst we feel it is unlikely that women would want a ‘negative experience’, our analysis aimed to capture any disconfirming data (see line 184 in the methods section). We have also been working closely with colleagues at the WHO to reframe maternity care within the parameters of good health and wellbeing rather than merely the absence of ill-health and/or clinical pathology. We appreciate that the development of a “positive experience” highlighted in Box 2 isn’t clearly described so have added some narrative to the introduction, the methods and the findings section to clarify this process.

 Last line of Introduction (112-115) – ‘The aim of the review is, therefore, to identify what matters to women in the postnatal period, in order to better understand how postnatal services can be optimally designed to deliver a positive experience to meet the needs of women, their families and their neonates’ 

 Last line of the Methods section (204-205) – ‘Key concepts relating to a positive postnatal experience were derived from the summary statement’. 

 Lines 257-259 in the Findings section – ‘Based on this statement we identified the key components of a positive postnatal experience to align with previous reviews of what matters to women during antenatal care [61] and intraprtum care [62]’

I was interested to see that nearly all the findings were rated as moderate or high confidence using CERQual, although the review question appears to be global, and only three of the studies relate to LMIC. I might therefore expect greater downgrading of “relevance” in the CERQual scores for these findings which don’t contain geographical limits. Typically, the CERQual scores are accompanied by a Summary of Findings table, which includes a brief explanation about how the overarching CERQual score was derived. This might improve the transparency of how the gradings were assigned (see table 4 https://journals.plos.org/plosmedicine/article?id=10.1371/journal.pmed.1001895).

 The CERQual Summary of Findings table is shown on page 4 of the S1 Appendix file. This includes the 4 CERQual domains (Methodological limitations, Adequacy of data, Coherence and Relevance) for each review finding with a narrative description for each domain and an explanation for downgrading where appropriate. As our included studies came from every continent (Europe, Australia, North & South America, Asia and Africa) we felt we had a relatively good geographical spread and each of the 3 African studies were conducted in different countries. 

Searches: No social science data base (such as ASSIA) was searched – is it possible that papers have been missed? Can a reference be provided for the qualitative research filter used in the searches on p.7? I am unclear of the rationale for requiring line 5 AND line 6.

 Yes – fair point. We included PsychInfo and CINAHL in our database searches which do pick up some of the social science literature. We also recognize that search strategies for qualitative studies may not retrieve ALL of the pertinent papers because of the unsophisticated indexing systems in many databases. That said, we acknowledge the reviewers comments and have added a sentence in the Discussion section (under limitations) to reflect this omission 

Discussion section (Lines 400-402) – ‘Our searches were carried out in health and medical databases and it is possible that the inclusion of social science databases may have identified more papers relating to women’s needs during the postnatal period’

 With regard to the search strategy we are unable to provide a reference as the strategy was agreed by consensus amongst the review authors and chosen to reflect an efficient and pragmatic approach. However, we did use a recognized system (P E O) to identify the Population (Women – Line 3), Exposure (Postnatal Care – Lines 1 & 7) and Outcomes (Line 6) as well as qualitative descriptors to denote study type (Lines 2 & 4). Line 5 represents the study type and Line 6 reflects the outcomes. There may be some overlap between these two lines but, in our experience, qualitative outcome measures (views, experiences, etc;) don’t always correlate with qualitative descriptors. 

Minor points: It would be good to see data analysis method, as well as collection method, recorded in Table 1.

 A brief description of the data analysis method for each included study is highlighted on page 1 of the S1 Appendix file. 

I’d be interested in the authors’ reflections on the appropriateness of using google translate in maintaining conceptual and idiomatic meaning for qualitative analysis, although this may be outside the paper’s scope!

 We acknowledge that conceptual translation between languages and cultures is an issue in both qualitative and quantitative research. *Regmi et al (2010) discuss the issues of translation (a direct and literal word‐for‐word process) and transliteration (a process of translating meaning which may not be word‐for‐word) in undertaking qualitative research in different language and cultural groups. They use the term 'elegant free translation', from **Birbili (2000) which is an approach that in Birbili's analysis can help the reader to 'know what is going on' even if it is less faithful to the original text. Regmi et al (2010) see this as "a process involving transcription of only the key themes or few quotes, putting in the context". They recognise that this risks the loss of some precision and meaning, but that it is a pragmatic solution to the complexity and resource demands of full translation in primary qualitative research. Given that the current review did not aim to be philosophically phenomenological, and that the data we were using (published in English or any other language) was at the level of author themes, selected quotes, and author interpretations of their primary data, we took the pragmatic decision to use the 'elegant free translation' approach to the transliteration of our included studies, rather than translating them word‐for‐word. For Latin based languages we find that “Google Translate” does a reasonable job in providing an elegant free translation

We can add some or all of this text to the manuscript by way of explanation.

*Regmi K, Naidoo J, Pilkington P. Understanding the processes of translation and transliteration in qualitative research. International Journal of Qualitative Methods 2010;9(1):16‐26.

**Birbili M. Translating from one language to another. Social Research Update 2000;31:1‐7.

There are some inconsistencies of language to iron out – for example, this paper is referred to as a “systematic scoping review” on line 401, which I think undersells the synthesis. 

 Yes. Fair point. We have changed the title to “a meta-synthesis of qualitative studies” to better reflect the methodological approach

I also suggest that the language of “overarching themes” rather than “summary themes” is kept throughout as the latter doesn’t capture the conceptual novelty of these themes. 

 Thank you. In line with the revised description of the methodology we have changed this term to “analytical themes” and have used this term consistently

---

## [Editor Report · Decision Letter 1]

24 Mar 2020

What matters to women in the postnatal period: A meta-synthesis of qualitative studies

PONE-D-20-01368R1

Dear Dr. Finlayson,

We are pleased to inform you that your manuscript has been judged scientifically suitable for publication and will be formally accepted for publication once it complies with all outstanding technical requirements.

With kind regards,

Christine E East

Academic Editor

PLOS ONE
---

## [Editor Report · Acceptance letter]

27 Mar 2020

PONE-D-20-01368R1 

What matters to women in the postnatal period: A meta-synthesis of qualitative studies. 

Dear Dr. Finlayson:

I am pleased to inform you that your manuscript has been deemed suitable for publication in PLOS ONE. Congratulations! Your manuscript is now with our production department. 

With kind regards,

on behalf of

Dr. Christine E East 

Academic Editor

PLOS ONE